# Health promotion interventions for the control of hypertension in Africa, a systematic scoping review from 2011 to 2021

Jinhee Shin[1], Kennedy Diema Konlan[1,2]*, Eugenia Mensah[3]

1 Mo-im Kim Nursing Research Institute, College of Nursing, Yonsei University, Yonsei-ro, Seodaemun-gu, Seoul, Korea, 2 Department of Public Health Nursing, School of Nursing and Midwifery, University of Health and Allied Sciences, Ho, Ghana, 3 War Memorial Hospital, Navrongo, Upper East Region, Ghana

* dkkonlan@uhas.edu.gh

**Data Availability Statement:** All relevant data are within the paper and its Supporting Information files.

## Abstract

### Background

A proportion of hypertension patients live in developing countries with low awareness, poor control capabilities, and limited health resources. Prevention and control of hypertension can be achieved by applying both targeted and population-based health promotion interventions. This study synthesised the health promotion interventions for the control of hypertension in Africa.

### Methods

An in-depth search of PubMed, CINAHL, EMBASE, Cochrane library, web of science, google scholar yielded 646 titles and 615 after duplicates were removed. Full text (112) was screened, and ten articles were selected. The data analysis method was thematic analysis through the incorporation of convergent synthesis. The major sub-themes that were identified were reduction in the prevalence of hypertension, increase in knowledge, impact and feasibility, role in the reduction of risk factors, and the cost associated with health promotion interventions.

### Results

Health promotion interventions led to a remarkable decrease in the prevalence of hypertension, increased knowledge and awareness in the intervention compared to the control groups. Community-based interventions were noted to have a positive impact on people's adoption of measures to reduce risk or identify early symptoms of hypertension. There was a significant relationship for the reduction in salt consumption, smoking, alcohol use, and increased physical activity after the administration of an intervention. Interventions using community health workers were cost-effective.

**Funding:** The authors received no specific funding for this work.

**Competing interests:** The authors have declared that no competing interests exist.

## Conclusion

To sustain health promotion interventions and achieve control of hypertension especially in the long term, interventions must be culturally friendly and incorporate locally available resources in Africa.

## Introduction

Worldwide, hypertension causes significant morbidity and mortality, contributing to 57 million (3.7% total) disability-adjusted life years and 7.5 million (12.8%) premature deaths annually [1]. The incidence of hypertension among Africans is noted to be higher than Caucasian populations [2,3], and remains an emerging public health problem especially in developing countries. Most hypertension patients (639 million) live in developing countries where they are faced with low awareness, poor control capabilities, and limited health resources [4,5]. This high prevalence and poor control of hypertension are important factors in the increasing prevalence of cardiovascular disease especially among Africans. Poor hypertension control is noted to lead to increasing prevalence of haemorrhagic and ischaemic stroke, ischaemic heart disease, cardiovascular disease, heart failure and other peripheral heart diseases [6,7].

Hypertension is a complex polygenic disorder that is influenced by combinations of genetic, environmental, socio-economic and demographic factors [8]. Genetic factors are noted to be influenced by the environment as modifiable epigenetic factors are known to be inherited over several generations [9,10]. Although the genetic predisposition cannot be modified, the risk of hypertension can be lowered by modifying key environmental and lifestyle factors. The important factors that increase hypertension prevalence in childhood and early adulthood are weight gain leading to obesity, excessive sodium, inadequate potassium intake, insufficient physical activity, and consumption of alcohol [9–11]. Key improvements can be made through individual adoption of positive behaviour that minimises the risk of hypertension. These can be attained through sustained implementation of lifestyle modifications that limits the risk associated with hypertension [12] and can be achieved through the implementation of health promotion interventions that ensure sustained control.

Prevention and control of hypertension can be achieved by applying both targeted and a population-based strategies. The targeted approach is a traditional strategy used in clinical practice, which seeks to reduce high blood pressure among clinical patients. The strategy that uses population-based approach is derived from public health mass environmental control experiences that do not specifically target a particular set of the population [13]. The goal in this strategy is to have little reduction in blood pressure within the population as these may have a downward shift in the population's overall risk and prevalence [2,8,10,13]. It is generally believed that the population-based approaches offer greater potential for preventing cardiovascular diseases than the targeted strategies [13,14]. This is based on the principle that many people exposed to small cardiovascular disease risk may result in more cases than few people exposed to various risks [13]. It is noted that reducing the diastolic blood pressure in the general population by 2mmHg would be expected to reduce the incidence of hypertension (17%), stroke (14%), and coronary artery disease by 6% [6,15]. However, both strategies may use the same interventions, as they are complementary and mutually reinforcing emphasizing the imperative to institute health promotion interventions that are key in the control of hypertension in the at-risk population and among patients.

Research over the decades has implemented several strategies in Africa with the goal of identifying acceptable, culturally friendly, feasible, and cost-efficient means for the control of

hypertension. These tested strategies are noted to be geographically sporadic, uncoordinated, and have produced diverse outcomes or impacts. It is therefore imperative to ensure synthesis and collation of these studies in a single document to ensure easy implementation for the control of hypertension. This study synthesised health promotion interventions for the control of hypertension in Africa.

## Materials and methods

### Design

Primary research articles published between 2011 to 2021 were reviewed and reported using the Preferred Reporting Items for Systematic Reviews and Meta-Analysis (PRISMA) framework [16–18]. The time frame of 2011 to 2021 was chosen to critically examine the health promotion interventions that are adopted for the prevention of the risk of hypertension in Africa. This review was conducted from April to August 2021.

### Search strategy

We searched six (PubMed, CINAHL, EMBASE, Cochrane library, web of science, google scholar) electronic databases for eligible studies after making scoping searches through manual search guided by the reference list of the selected studies. The key words were first searched in Pubmed, and the corresponding medical subject heading of indexed keywords were identified. The medical subject headings (MeSH) for index words and free text search for non-indexed words were searched by combining the appropriate Boolean operators in the various electronic databases using the advanced search option. In google scholar the first three pages comprising 150 titles were searched manually and appropriate manuscripts were selected. The population, intervention, comparison, and outcome (PICO) framework was integrated in PRISMA in searching, screening, and selecting eligible studies. The population were adolescents and young adults, the intervention was any health promotion intervention, comparisons were not clearly defined, and the outcome was hypertension control [16–18]. The MeSH terms and the free text words and phrases that were searched with the appropriate Boolean operators were (hypertension OR high blood pressure OR elevated blood pressure OR HTN or hypertensive) AND (health promotion OR health education OR patient education) AND (adolescents OR teenagers OR young adults OR teen OR youth). The articles were screened for only African, English-based articles from 2011 through to April 2021.

### Search results

In searching from the six electronic databases, 646 titles were identified from using the keywords and 615 titles after duplicates removal. After filtering for only English, published in Africa and within 2011 to 2021, only 615 titles were eligible for abstract and full-text screening. The titles for all these 615 titles were read and screened independently by two of the researchers. Titles that did not focus on health promotion intervention were excluded for full-text screening. The total number of articles selected from this round of screening was 112. abstracts. The full-text of these selected abstracts was screened, and ten articles were deemed to be eligible for this study.

### Inclusion and exclusions criteria

The selection of each study depended on predefined inclusion and exclusion criteria. The inclusion criteria took into consideration the following: article focused on a health promotion intervention, study conducted in Africa, published in English and between 2011 through to

2021. The exclusion criteria included articles that measured prevalence and incidence of hypertension, identified comorbidities, and focused on other variables than health promotion intervention.

### Data collection and extraction

To extract data, a matrix was first developed, discussed, and agreed on by all the researchers. The matrix guided the way the data was extracted. All the researchers independently extracted data using the matrix. The extracted data were compared and where there was a discrepancy, it was resolved through discussion and consensus. The major variables that were contained in the matrix included. Authors and year of publication, study design, setting, population and sample, data analysis and measurement tool for outcome variables. Other variables extracted were the intervention provider, duration of intervention, name of health promotion intervention and the key findings.

### Quality appraisal

The tool used for quality appraisal is the Mixed Methods Appraisal Tool (MMAT) version 2018. Two researchers independently appraised the quality of each study as suggested by Hong et al., 2018 [19]. The appraised data were then compared and where there were discrepancies, it was resolved by consensus. Where a consensus could not be reached, the two researchers consulted a third person, and the majority decision prevailed. The MMAT tool contains methodological criteria for the appraisal of quantitative, qualitative, and mixed methods studies as well as for interventional and experimental studies. The section that was appropriate for the appraisal of the studies was the part that pertains to randomised control trials, non-randomised control trials and mixed methods study. The MMAT tool allowed for screening of each research work by confirming a response of affirmation or disagreement. If there were clear research questions and if the data collection method allowed for the addressing of the research questions. All the studies were affirmative to these screening questions.

The appraisal questions for the section that entailed in randomised control trials section were five and include the appropriateness of randomisation, comparability of groups at baseline, completeness of outcome data, the blindness of assessors and participants adhered to assigned interventions. Only one study was evaluated in this category and had a negative response in respect of the questions that pertained to the blinding of the assessors in respect to the intervention [20]. Five questions were related to the non-randomised control section. These criteria include representativeness of the sample to the population, appropriateness of measurement of both the outcome and the intervention, completeness of outcome data, accounting for confounders in the design and analysis, and whether the interventions were administered as expected. The appraisal showed that some studies did not meet the criteria pertaining to the representativeness of study participants to the population [21–24], presence of complete outcome data [25], accounting for confounders in the design and analysis [22,24,25] and whether an intervention was administered [24]. One of the studies used a mixed-method study design and met all the criteria that include the adequacy of rationale for mixed-method study, appropriate integration of quantitative and qualitative data, divergence, and inconsistencies between quantitative and qualitative results, and that the different levels of the study adhered to the quality criteria of each tradition of the methods used. It did not however meet the criteria for different components of the study effectively integrated to answer the research question [26]. One of the studies was a quantitative study [25] and met all the criteria questions that are stipulated for this section.

## Data analysis

The data analysis method that was adopted is convergent synthesis. The convergent synthesis method was used because of the diversity of study designs that were adopted by the primary studies [27,28]. Prior to the synthesis, there was identification and description of the various health promotion interventions that were used for the prevention of hypertension. To use a convergent synthesis analysis method, the study findings were translated into descriptive qualitative sentences [28]. There was then a purposeful collation and integration of the findings [29] into themes from subthemes that were developed from the codes generated. In the views of Pluye and Hong, while conducting this type of synthesis, the various items identified must be integrated into subthemes and similar or related sub-themes collated to form broad umbrella themes [29]. The integration of quantitative and qualitative findings mainly occurred in the coding and development of sub-themes stage [27]. These themes were then described to have meanings that are beyond the originally identified items and hence allow for interpretation and critical analysis. No subgroup analysis and test of the robustness of study finding was conducted as this study was mainly aimed to be a narrative synthesis.

## Results

There was an in-depth search of six (PubMed, CINAHL, EMBASE, Cochrane library, web of science, google scholar) electronic databases that yielded 646 titles and 615 after duplicates were removed, as shown in Fig 1. The titles, abstracts, and full text (112) were then screened, and ten articles were identified as appropriate for this study.

### Study characteristics

The study designs adopted were survey [21,25], quasi-experimental [30–32], Cohort [22,24,25], mixed methods [26], randomised control [20], and exploratory uncontrolled pre–post intervention [23] as shown in Table 1. The studies were conducted in the Faculty of Pharmacy at Rhodes University [21], the Gugulethu township of Cape Town [26], Khayelitsha [24] all in South Africa [22,25], Sousse in Tunisia [30], Afon and Ajasse Ipo districts in Kwara State in Nigeria [31], three community pharmacies in the Ashanti Region of Ghana [23], and the slums of Korogocho and Viwandani in Nairobi [30]. The target participants for the health promotion interventions had a minimum age of eleven [21] and a maximum of sixty-five years [30] and included both sexes. The sampling methods adopted were the convenience [21,23,24,26,30], stratified convenience [25], random [30], stratified 2-degree random probability by geographic areas [31], and the Markov model used as a tool for sampling with the focus on age variability of cardiovascular diseases [22]. The study duration for each intervention study ranged from a minimum of a day health education programme [21] to a maximum of 36 months health promotion for behaviour change [30]. The other studies were 3months [25], 4 months [20,26], 5 months [23], 9 to 10 months [25], 12 months [22], 18months [32] and 24 months [24,31] health promotion interventions.

### Reduction in prevalence of hypertension after health promotion interventions

Health promotion interventions were noted to have a positive impact on the prevalence of hypertension [30–32] as shown in key findings in Table 2. The health promotion interventions led to a remarkable decrease in the prevalence of hypertension in the intervention compared to the control groups [30,31]. It was reported that the prevalence of hypertension decreased in the treatment group globally from 37.3% to 33.7% [30]. After stratification for age, for

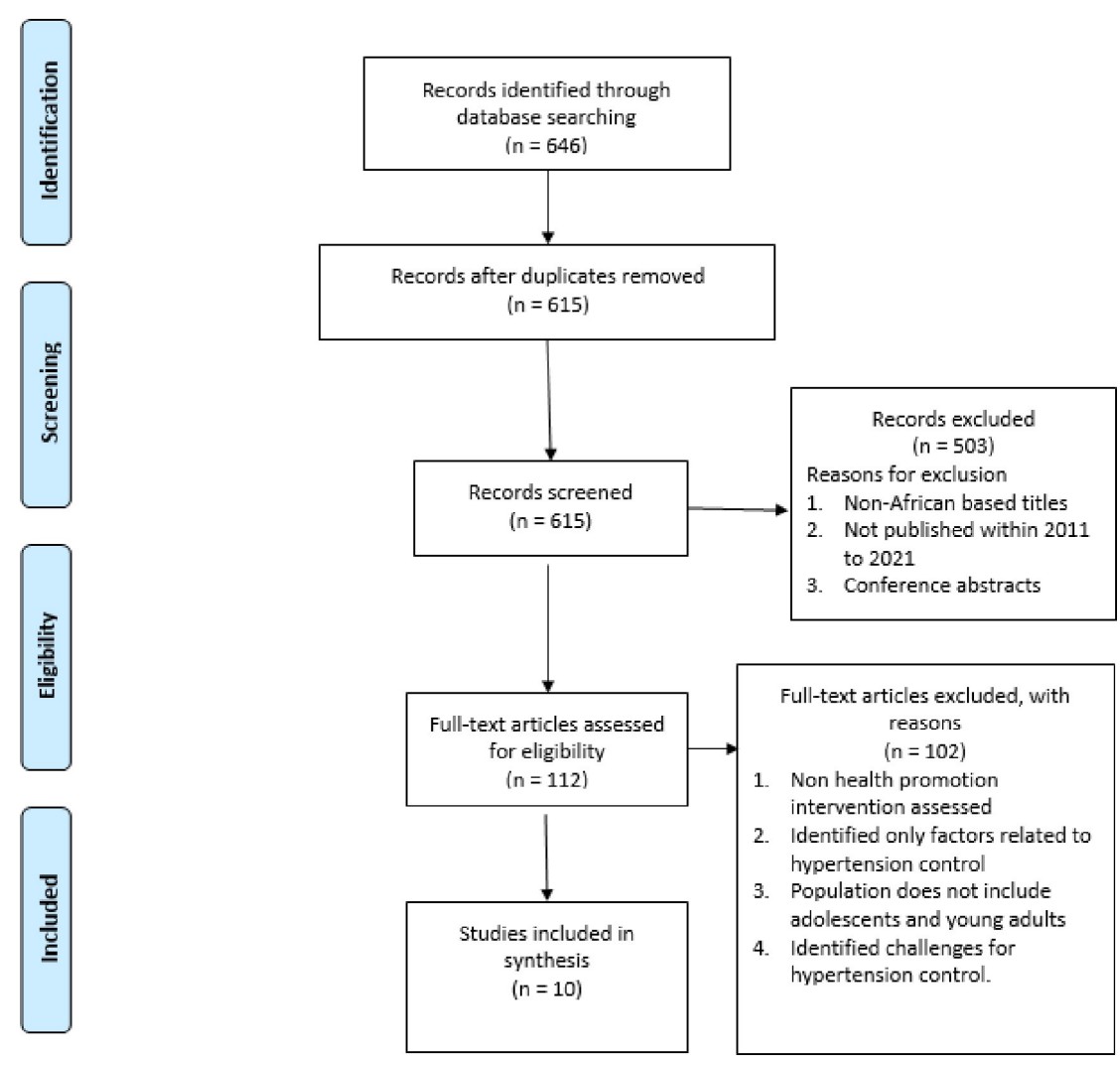

**Fig 1. PRISMA flow charts.**

participants younger than 40 years old, a significant decrease in the prevalence of hypertension from 22.8% to 16.2% in the intervention group and 14% to 15.4% in the control group was also noted [30]. A significant decrease in the prevalence of hypertension from 31.4% to 26% was observed among nonobese participants in the intervention group [30].

It was also shown that health promotion interventions were noted to improve the intervention group's ability to control blood pressure [30–32]. The number of respondents with controlled blood pressure increased from 3.0% to 38.8% in the program area, and a lower increase (4.0% to 26.1%) rate was noted in the control group [31]. In instances where there were improvements for hypertension prevalence in both the intervention and the control groups, the changes in the intervention groups were noted to be remarkable [30,32]. Mean blood pressure was said to reduce remarkably in intervention groups than in the control groups [31,32]. There was a significant reduction in mean SBP between baseline and end-line measurements in the intervention (2.75 mmHg) than the control (1.67 mmHg) groups [32]. It was statistically significant that systolic blood pressure decreased by 10.41 mmHg in the intervention, representing a 5.24 mm Hg greater reduction compared with in the control, which showed a

**Table 1. Distributing of study characteristics.**

| Citation | Design | Settings | Sampling and size | Analysis | Outcome variable | Measurement tool |
|---|---|---|---|---|---|---|
| Srinivas et al., 2015 [21] | Survey | South Africa Rhodes University, faculty | • 203 Scifest participants aged 11 -15years<br>• Convenience sampling | t-tests and ANOVA procedures | Hypertension Knowledge | Self-developed pre and post intervention quizzes (tool) |
| Sahli et al., 2016 [30] | Quasi experimental study | Tunisia, Sousse | • 2000 adults based on census data of people aged 18 to 65<br>• Random sampling | the binary logistic regression model Chi-square test | Medical history, attitudes, and beliefs | Self-developed in collaboration with community intervention |
| Wentzel viljeon et al., 2017 [25] | Cohort study | Gauteng, Eastern Cape, Kwa Zulu-Natal, South Africa | • 550 at baseline and 477 at follow-up of 18- 55years black women.<br>• Convenience stratified sampling | Multinomial regression models | Blood pressure | Self-developed baseline and follow up questionnaire |
| Gaziano et al., 2014 [22] | Cohort Study | South Africa | • 27% study people aged 25 to 74<br>• Markov model as a tool for sampling with the focus on age-varying | Probabilistic sensitivity analysis using variable models | Blood pressure and cholesterol level | Self-developed questionnaire and WHO choice |
| Hendrik et al., 2014 [31] | Quasi-experimental design | Nigeria, Afon, and Ajasse Ipo districts in Kwara State | • 1500 households<br>• A stratified 2-degree random probability from a random sample of geographic areas | Descriptive statistics | Blood pressure | Consecutive household surveys |
| Hacking et al., 2016 [26] | A mixed-methods approach | Gugulethu, Cape Town, South Africa | • 223 picked from a cohort chronic hypertension outpatient clinic<br>• Randomly grouped under control and intervention groups | Fisher's exact tests 2 sample t-tests | Knowledge and self-reported behaviour change | Questionnaire on self-reported behaviour changes |
| Rampamba et al., 2019 [20] | Quasi-experimental design | South Africa | • 253 patients through stratified random sampling (138 in intervention and 115 in the control group). | Fisher's exact tests pre and post-behaviour change | Knowledge of hypertension | Self-developed questionnaire |
| Marfo et al., 2016 [23] | Exploratory pre–post intervention | 3 community pharmacies, Ashanti Region Ghana | • 170 aged 45 years and above with obesity, diabetes, or smoking<br>• Convenience (visited the 3 pharmacies for a refill of medication) | The McNemar's Chi-square test | Blood pressure BMI prevalence | Pre–post intervention tool |
| van de Vijver et al., 2016 [32] | Prospective intervention study | Slums of Korogocho and Viwandani in Nairobi | • 1,233 participants aged above 35 years<br>• Convenience sampling | Logistic and linear regression | Awareness of hypertension and cardiovascular disease risk | Self-developed questionnaire |
| Puoane et al., 2012 [24] | Review: Cohort Study design | Khayelitsha, South Africa | • 76 participants: only 22 were regular attendees, and two years later, the number increased to 30<br>• Facilitation of health club | Descriptive statistics | Blood pressure BMI | None |

decrease of 5.17 mmHg only. Diastolic blood pressure decreased by 4.27 mmHg in the intervention, a 2.16 mmHg greater reduction compared with the control, where blood pressure decreased by 2.11 mmHg [31].

## Knowledge increase after health promotion intervention

Health promotion interventions that sought to increase community knowledge on hypertension yielded positive outcomes as knowledge levels were noted to increase [21,23,25]. In a post-intervention quiz, there was a significant increase in the scores from 78.2 to 85.6% in

**Table 2. Distribution of key findings.**

| Citation | Intervention provider | Main health promotion intervention | Key findings |
|---|---|---|---|
| Srinivas et al., 2015 [21] | 5 fourth-year pharmacy students and faculty | • Health education information and materials were provided to the participants<br>• There was pre and post-intervention quiz | • Averagely the participants demonstrated a level of knowledge on the disease condition during the pre-intervention stage.<br>• There is a slight increase in the knowledge levels between the government school and non-state- funded schools. |
| Sahli et al., 2016 [30] | Physicians, paramedics, nutritionists, and a psychologist. | • The main intervention was streamlined into the promotion of health education on healthy living and lifestyle modifications<br>• Open sensitization, educational flyers, and mass media interventions were done | • The study showed a significant decrease in hypertension among the nonobese participants in the intervention group.<br>• The feasibility and effectiveness of the intervention depicted a reduced prevalence of hypertension in developing countries. |
| Wentzel viljeon et al., 2017 [25] | Salt reduction stakeholders formed an advocacy group called Salt Watch. | • Health education awareness on the impact of increased salt intake<br>• The awareness took the form of talks by medical practitioners, stand interactions, flyers, media education | • The study revealed a shift in behavioural intake of salt in black women.<br>• The health promotion intervention yields a better result in the reduction of salt intake by most households. |
| Gaziano et al., 2014 [22] | 6 community health nurses and nurse coordinators | • The health workers were assigned to 6 home visits per day based on a population density of approximately 2500 adults/5 km2<br>• Provided health education and hypertension measurement | • Investing in community healthcare worker intervention was cost-saving and reduced mortalities. Community healthcare workers have an impact on chronic diseases leading to improved blood pressure control. |
| Hendrik et al., 2014 [31] | Health professionals | • Blood pressure was taken three times on the upper arm<br>• An educational leaflet on hypertension was given to the household<br>• Health education was given to each household | • Uncontrolled hypertension in baseline had controlled blood pressure in 2011 without reporting any medication or lifestyle intervention to hypertension.<br>• Depicted health insurance programs that covered the costs of care for patients and improved the quality of health care facilities. |
| Hacking et al., 2016 [26] | Health promoters and staff | • Administered a pre-intervention multiple-choice questionnaire<br>• SMS constituted health tips and cues to be taken seriously concerning the disease condition<br>• Focus group discussion on health tips received from the Short Message Service | • Short Message Service was seen as an effective and a positive model in lifestyle modification; however, there was little significance in the content message. |
| Rampamba et al., 2019 [20] | Pharmacist | • 15–30 minutes Patient counselling and education: hypertension information diary for daily use, correct use of the diary | • Improvement in knowledge regarding hypertension in the intervention group (34.7%, P < 0.001).<br>• In the intervention group, improvement in the knowledge that systolic and diastolic blood pressure are important in controlling hypertension (9.1%).<br>• Patients (40.0%) in the intervention group versus the control group (17.9%) had adequate knowledge ($\geq$75% correct answers) about hypertension and its management |
| Marfo et al., 2016 [23] | Five pharmacists and five medicine counter assistants | • Health awareness discussions, educational leaflets were provided<br>• Focused on non-pharmacological measures like reduction in alcohol intake, frequent exercise, and maintaining a healthy diet | • Good benefits of health promotion intervention led to changed ideas and lifestyle modification.<br>• Pharmacist-led hypertension preventative services were seen as feasible and acceptable by the sampled number. |
| van de Vijver et al., 2016 [32] | Private and public health care workers, various stakeholders | • Awareness campaigns, household visits for screening, referral, and treatment of people with hypertension<br>• Promoting long-term retention in care | • Found significant declines in systolic blood pressure over time in both intervention and control groups.<br>• No additional effect of a community-based intervention involving awareness campaigns, screening, referral, and treatment. |
| Puoane et al., 2012 [24] | Health clinics staff members, community health workers | • 4 fun walks, two diabetes workshops to create awareness<br>• Nutrition education sessions were held once a month and cooking demonstrations were incorporated into the sessions | • Two years after the intervention, there was a reduction in the number of participants who were obese (i.e., BMI > 30 kg/m2).<br>• Overweight and obesity remain a problem in this population. |

hypertension knowledge among those that received a health promotion intervention [21]. During a post-intervention survey, it was also noted that 40% of the participants reported having heard, read, or seen any food and/or health-related advertisement campaign in the last few months, compared to less than 20% at baseline, across all age and LSM groups [25]. Participants' awareness of having hypertension in the intervention group was noted to be higher than in the control group [32]. In Ghana, people among the intervention group who were referred to the hospital because they had higher blood pressure (>140/90) did not need to be put on medication [23]. Most of the respondents with hypertension were unaware of their status during the baseline survey but showed significant awareness upon implementation of the intervention [31]. In another health promotion intervention, it was observed that overall knowledge about blood pressure and hypertension increased among those who received treatment [20,26]. It was noted that 40.0% from the intervention group and 17.9% in the control group showed improved knowledge on hypertension [20]. Health promotion on text messaging to hypertension participants on medication adherence was also noted to have a remarkable impact as the treatment arm demonstrated a significantly higher knowledge for an extended duration [26]. In instances where the intervention group was given diaries to use, 97.7% showed it benefited them to remember their medication and clinic appointment [20]. After a media campaign, participants were identified to adopt positive lifestyle modifications (weight loss and no salt or alcohol intake) that reduced their risk of hypertension [25]. After a media campaign for the reduction in salt intake, 77.8% reported that they had seen or heard the specific SaltWatch media campaign that included salt-related health information on TV and radio [25].

## Feasibility of health promotion interventions and impact

Community-led health promotion interventions were noted to have a positive impact on people's adoption of measures to reduce risk or identify early the symptoms of hypertension [22,23,31,32]. It was noted that the community pharmacy is a feasible setting for screening and detection of hypertension if the right structures are put in place [23]. The use of this intervention strategy is also appropriate due to easy accessibility for providing information on lifestyle practices to prevent hypertension [23]. During community intervention programs, newly identified people who have hypertension are referred to the health facility to seek and use professional care [31]. The use of these intervention strategies has led to an increase in the antihypertensive drug treatment from 4.6% to 13.1% among those that were screened- the intervention group [31].

## Health promotion interventions reduced hypertension risk factors

Most of the indicators of knowledge, attitudes, and behaviour change showed a statistically significant relationship for the reduction in salt consumption, smoking, alcohol use, and increased physical activity after the administration of an intervention [23,25]. Significantly more participants reported that they were taking steps to control salt intake especially adding salt while cooking and at the table [25]. Given the message that was communicated during the health promotion intervention, the participants could readily remember the key messages that are likely to improve the chance of behaviour change. The most frequently recalled messages were that "too much salt is bad for your health" followed by "you should eat less salt", these made participants who thought they had consumed the right amount of salt-reduced salt intake [25]. Among patients with hypertension in the control group, smoking and alcohol use were also reduced significantly [32]. It was also noted in Ghana that physical activity levels increase significantly among intervention than in the control groups [23]. It was also noted

that there was a significant decrease in the numbers of those reporting inadequate physical activity among the intervention compared with the control group at the population level and among hypertension people at baseline [32].

## Cost of health promotion interventions

Community health worker intervention was noted to be cost-effective as it led to a remarkable reduction in the cost of care for hypertension patients [22]. Once the annual cost per patient was below $6.50, the community health worker intervention became "cost-saving" because it saved costs and increased life expectancy, especially when the blood pressure reduction was above 4.98 mmHg [22]. After text messaging, the intervention group had positive increases in self-reported behaviour changes [26]. Health promotion interventions were also noted to produce an improvement in health insurance coverage as the intervention group had a 40.1% increase and the control had less than a percentage point [31]. Self-reported general use of health care resources increased in the program area and decreased in the control area [31].

## Discussion

This systematic review synthesis the health promotion interventions that are critical in the control of hypertension in Africa. It is important to note that the major determinants of hypertension can be categorised into genetic or epigenetic and environmental or social factors that interact in a complex iterative fashion to increase an individual's risk and the ability to control hypertension. Hypertension health promotion interventions are mostly targeted to those factors that can be altered through individual efforts-largely referred to as modifiable risk factors [33–36]. These targets of health promotion intervention incorporate those environmental and social determinants of health that include lifestyle factors like heart-healthy diet [37,38], reduction in sodium and adequate potassium [35–38], increased physical activity [37], reduction in overweight and obesity [39–42] as well as increased knowledge on hypertension risk factors [11,35–38,43]. The specific target of these modifiable risk factors, especially among the entire population, has been shown in this study to be critical if significant gains are going to be made in the total control of hypertension. Specific health promotion interventions that are reviewed showed significant positive improvement in knowledge and people's adoption of behaviours that reduce the risk associated with hypertension.

To ensure sustained hypertension control among those diagnosed and reduce the incidence, several barriers are identified to be implicated. These barriers included cultural norms, insufficient attention to education, lack of resources for interventions for hypertension control. Other barriers associated with population-based hypertension control included poor health education, lack of physical activity culture and space to engage in same, urbanisation and its attendant increased in restaurants and the consumption of fast foods rich in calorie and fat, consumption of large amounts of sodium and lower potassium, and inadequate information on how to control hypertension [11,43,44]. Health promotion interventions that specifically target mitigation or the elimination of these barriers have been shown through the various studies in Africa to be cardinal. It is important that in a resource-limited setting like Africa, health promotion interventions specifically target these barriers and identify means to mitigate the same. Other factors incorporating wealth and income levels and social determinants like employment, access to health care, social inequalities are noted to influence individual ability to adapt to measures that prevent hypertension [45,46]. These factors are identified to hinder the early detection, awareness creation, control, and management of hypertension in Africa. It is therefore imperative that multi-pronged approaches are adopted to target all

populations (at work, school, and industries) and not only those at risk. In developed countries, there have been many health promotions programs for hypertensive patients to change modifiable factors [45–47].

There are several limitations in implementing health promotion programs in developing countries compared to developed countries. Health promotion interventions are mostly messages that are communicated and, in some instances, will require the extensive reading of health information material. In Africa, literacy levels remain low, coupled with the relative lack of a common language that is usually locally accepted and understood by all. The contents of most health education programs in developing countries are often difficult to read and understand by most people because of relatively low educational levels [48]. This makes the training method for health promotion interventions to be rather tedious and inefficient, and hence the implementation of health promotion interventions in these low resource settings relatively difficult. However, the use of culturally friendly, easily understandable, and the use of local resources was seen as one of the best means of health promotions interventions and has the propensity to mitigate the difficulty associated with language. It was realised that the use of community-based pharmacy, health education granted in local languages, and use of next of keen as reminders on medication adherence has been keen in early detection, increased knowledge, and appropriate medication adherence among hypertension patients, respectively. There has been increasing interest in using diverse strategies for measures that can curtail hypertension prevalence. The use of telecommunication is gaining widespread popularity in African countries and leveraging of such means promises to be one positive means of health promotion for persons at risk. The use of cell phones and short messaging services, and mobile notifications have been shown to have positive effects in several Human immunodeficiency virus infection intervention studies [49,50] and hypertension patients [26] in Africa. In response to the changing African environment, useful and accessible methods of disseminating health-promoting knowledge, especially for the prevention of chronic diseases like hypertension, must be developed and implemented.

Several studies have shown health promotion interventions for people with hypertension particularly yield positive outcomes. This has even been the case over two decades ago when it was reported that a sustained 5years campaign for the implementation of measures to reduce the incidence of hypertension resulted in a 2.9% and 1.5% reduction in prevalence among men and women, respectively [51]. Similarly, other health promotion interventions that target physical activity resulted in a significant decrease in the prevalence of hypertension among the intervention group after five years of implementation [52]. Various health promotion interventions have resulted in a significant decrease in the number of obese participants, increased physical activity, and decreased salt intake [23,42], which are particular risk factors to hypertension [7,52,53].

It is also important that significant health promotions intervention that target hypertension risk factors focus on salt intake, a significant risk factor for hypertension [54]. Since reducing salt intake reduces blood pressure [53], it is often used as an intervention strategy. Salt reduction strategies based on improving individual and group health, increasing awareness, and changing behaviour should be relatively easy to implement and have a high probability of hypertension risk reduction. A health education program (six months of education) on the harmful health of high salt intake provided by a community healthcare provider to residents has lowered the population's prevalence of blood pressure with an average reduction of 2.5/3.9 mmHg in the intervention group [55]. For a salt reduction of less than 3g, the mean population systolic blood pressure decreased by 1.3mmHg. It must be noted that these are cost-effective and useful interventions that can produce tremendous results for poor resource settings.

### Strengths and limitations

This study provides a comprehensive overview of the health promotions interventions that are used for the control of hypertension in Africa. It is important to note that all the researchers worked as a team in all the phases of this study, and where there was a disagreement, a consensus was built. This reduced the likelihood of subjectivity that is usually associated with study selection, data extraction, and analysis in systematic reviews. The study is not without limitations as only English-based articles were included in the study, creating the possibility of some salient articles in other languages left out. The protocol for this study did not receive prior registration. Also, the quality assessment of the included studies was minimal as it was largely limited to assessment for only the risk of bias.

## Conclusions

This study showed the role of health promotion interventions in the control of hypertension in poor settings in Africa. It was realized that health promotion interventions that focus on increasing education, information dissemination, and promoting behaviour change were seen as useful in the control of the entire hypertension incidence and prevalence. Interventions that use local resources and are largely community-based also showed positive health outcomes. It is imperative that to sustain health promotion interventions and achieve control of hypertension, especially in the long term, interventions must be culturally friendly and incorporate locally available resources. It is also noted that health promotion interventions that are coupled with the increase in knowledge were seen to improve people's tendency to be healthy and to screen for early detection and treatment of hypertension. These types of intervention need to be further tested in various cultures of Africa and to ensure sustained prevention of hypertension risk factors.

## Supporting information

**S1 Checklist. PRISMA 2020 checklist.**
(DOCX)

**S1 File. MMAT appraisal of individual studies.**
(DOCX)

## Author Contributions

**Conceptualization:** Jinhee Shin, Kennedy Diema Konlan.

**Data curation:** Kennedy Diema Konlan, Eugenia Mensah.

**Formal analysis:** Jinhee Shin, Kennedy Diema Konlan.

**Investigation:** Eugenia Mensah.

**Methodology:** Jinhee Shin, Kennedy Diema Konlan, Eugenia Mensah.

**Project administration:** Kennedy Diema Konlan.

**Writing – original draft:** Kennedy Diema Konlan, Eugenia Mensah.

**Writing – review & editing:** Jinhee Shin, Kennedy Diema Konlan.

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
