## [Decision Letter · Decision Letter 0]

8 Oct 2021

PONE-D-21-28474Health promotion interventions for the control of hypertension in Africa, a systematic scoping review from 2011 to 2021.PLOS ONE

Dear,

Thank you for submitting your manuscript to PLOS ONE. After careful consideration, we feel that it has merit but does not fully meet PLOS ONE’s publication criteria as it currently stands. Therefore, we invite you to submit a revised version of the manuscript that addresses the points raised during the review process.

We look forward to receiving your revised manuscript.

Kind regards,

Muhammad Shahzad Aslam, Ph.D.,M.Phil., Pharm-D

Academic Editor

PLOS ONE

Journal Requirements:

2. PLOS ONE requires systematic reviews to include a detailed analysis of the quality of each study included in the review. Please attach a Supplemental file of the results of the quality assessment for each individual study assessed, broken down into individual quality assessment measures. Please also discuss how results can be interpreted given the quality of the included studies.

3. Please include your tables as part of your main manuscript and remove the individual files. Please note that supplementary tables should remain as separate "supporting information" files.

Reviewers' comments:

Reviewer's Responses to Questions

**Comments to the Author**

1. Is the manuscript technically sound, and do the data support the conclusions?

Reviewer #1: Yes

Reviewer #2: Partly

2. Has the statistical analysis been performed appropriately and rigorously? 

Reviewer #1: Yes

Reviewer #2: N/A

3. Have the authors made all data underlying the findings in their manuscript fully available?

Reviewer #1: Yes

Reviewer #2: Yes

4. Is the manuscript presented in an intelligible fashion and written in standard English?

Reviewer #1: Yes

Reviewer #2: Yes

5. Review Comments to the Author

Reviewer #1: Thank your for your efforts in contributing to literature on health promotion interventions on hypertension. However, some general concerns need to some explanations.

1) Why the choice of convergent sythesis analysis?

2) Could you please explain how the convergent synthesis analysis method allowed for the findings into descriptive sentences?

3) Was there a sequence in the synthesis of the evidence?

4) Where did the integration of quantitative and or qualitative/ mixed-method evdence occur?

5) How did you appraise the quality of retained studies to check the trusthworthiness of included studies?

6) Was there a sequence in the synthesis of the evidence?

7) Strengths and limitations: "This study provides a complete overview of the health promotions interventions that are used

for the control of hypertension in Africa." This is an overstatement, as only papers in English were included in the study. This is too much of a generalisation. Please edit this sentence.

Reviewer #2: This manuscript addresses a very important public health issues globally and particularly the surge in hypertension prevalence in developing countries and Africa. The scoping review is thoroughly done with the search methodology and screening processes and the manuscript is presented in an intelligible fashion. There are, however, some issues that I recommend you address to help make the manuscript more understandable and meaningful to the lay readers outside of health promotion.

1. There are some grammatical issues that I think you need to address and this calls for editing of the entire manuscript. Use can use Grammarly to edit the manuscript and that will help greatly.

2. In the results section, you have integrated the various health promotion interventions in the presentation. I suggest that you itemized the ten papers that were included in the final review and analysis and what of interventions they were and in what settings. This will make it clearer for the reader.

3. You have made some statements facts in the manuscript that you need to provide references for. 'The contents of

most health education programs in developing countries are often difficult to read and

understand by most people because of relatively low educational levels.' This is an example. Please, ensure these kinds of statements have references or put them in the form of probable statements.

4. In the strength and weakness section, you made a statement that I think is sweeping and I urge you to be cautious about such statements. 'This study provides a complete overview of the health promotions interventions that are used

for the control of hypertension in Africa". Yo can state that you have made a comprehensive review but not a complete review.

Overall, this manuscript is very important and will help health promotion program developers and implement evidence-based interventions with greater chances of success.

6. PLOS authors have the option to publish the peer review history of their article (what does this mean?). If published, this will include your full peer review and any attached files.

Reviewer #1: No

Reviewer #2: No

---

## [Author Response · Author response to Decision Letter 0]

26 Oct 2021

Department of Public Health Nursing

School of Nursing and Midwifery

University of Health and Allied Sciences

Ho. Volta Region

October 2021

Dear Sir,

Authors’ response to the manuscript review (PONE-D-21-28474)

We are most grateful to the editor and the reviewers for spending your precious time evaluating our manuscript (PONE-D-21-28474). In this cover letter, we have provided a point-by-point response to the comments made by the reviewer. We generally agree with most of the comments and observations made by the reviewers and have made substantial revisions to the entire manuscript. 

POINT BY POINT RESPONSE TO THE SUGGESTIONS MADE BY THE REVIEWERS

Here we provide a point-by-point response to each reviewer's comments. 

Reviewer #1

Reviewers’ comments: Thank you for your efforts in contributing to the literature on health promotion interventions on hypertension. However, some general concerns need some explanations.

Authors’ Response: We are particularly grateful for the valuable time you spent reviewing and helping to improve this manuscript.

Reviewers’ comments: 1) Why the choice of convergent synthesis analysis?

Authors’ Response: The authors have provided the basis for the choice and use of the convergent synthesis design. We have also provided references that influence this choice to include those of evidence articles (Hong et al., 2017; Noyes et al., 2019).

Reviewers’ comments: 2) Could you please explain how the convergent synthesis analysis method allowed for the findings into descriptive sentences?

Authors’ Response: With the inspiration of what was described by (Hong et al., 2017; Noyes et al., 2019). We have included how the convergent synthesis design was adopted. 

Reviewers’ comments: 3) Was there a sequence in the synthesis of the evidence?

Authors’ Response: There was a sequence in the synthesis of the evidence. To do this the authors were mainly influenced by the views of Hong et al., 2017; Noyes et al., 2019; Pluye & Hong, 2014. The three authors above encourage first the development of codes, coalesce into subthemes, and then the main themes developed from it. 

Reviewers’ comments: 4) Where did the integration of quantitative and or qualitative/ mixed-method evidence occur?

Authors’ Response: All the findings were first of all translated into descriptive findings, then coded, similar codes coalesced into subthemes, and related subthemes integrated into the main themes that we presented. the integration of the data from various designs were conducted primarily in the coding stage and through the development of the subthemes. 

Reviewers’ comments: 5) How did you appraise the quality of retained studies to check the trustworthiness of included studies?

Authors’ Response: The include studies that were appraised using the MMAT quality appraisal tool as described by Hong et a., 2018. In the methodology section, we provided a summary of the appraisal results under the subheadings quality appraisal. We have also attached the full appraisal results as a supplementary file to this reviewer.

Reviewers’ comments: 6) Was there a sequence in the synthesis of the evidence?

Authors’ Response: This comment has been addressed in number three above. 

Reviewers’ comments: 7) Strengths and limitations: "This study provides a complete overview of the health promotions interventions that are used for the control of hypertension in Africa." This is an overstatement, as only papers in English were included in the study. This is too much of a generalization. Please edit this sentence.

Authors’ Response: The authors have made a revision of this statement to show that a comprehensive review was made and not a complete review. As we agree with the reviewer that only English-based studies were included in this review and hence cannot be described as a complete review of the findings in Africa. In line with this, we have made a substantial review of the statement. 

Reviewer #2

Reviewers’ comments: This manuscript addresses a very important public health issue globally and particularly the surge in hypertension prevalence in developing countries and Africa. The scoping review is thoroughly done with the search methodology and screening processes, and the manuscript is presented in an intelligible fashion. There are, however, some issues that I recommend you address to help make the manuscript more understandable and meaningful to the lay readers outside of health promotion.

Authors’ Response: We are particularly grateful for the valuable time you spent reading and making reviews of this manuscript. We do agree that given the nature of the trend of hypertension in Africa, it is important to identify and institute health promotion interventions in earnest to truncate the trend.

Reviewers’ comments: 1. There are some grammatical issues that I think you need to address, and this calls for editing of the entire manuscript. You can use Grammarly to edit the manuscript, and that will help greatly.

Authors’ Response: The entire manuscript was reviewed for grammatical errors and substantial corrections made. 

Reviewers’ comments: 2. In the results section, you have integrated the various health promotion interventions in the presentation. I suggest that you itemized the ten papers that were included in the final review and analysis and what interventions they were, and in what settings. This will make it clearer for the reader.

Authors’ Response: We submitted the summary table as supplementary. Following your comment, we have integrated the findings in the main manuscript.

Reviewers’ comments: 3. You have made some statements in the manuscript that you need to provide references for. 'The contents of most health education programs in developing countries are often difficult to read and understand by most people because of relatively low educational levels.' This is an example. Please, ensure these kinds of statements have references or put them in the form of probable statements.

Authors’ Response: The researchers have taken note of this comment and have made revisions in the manuscript to reflect this view.

Reviewers’ comments: 4. In the strength and weakness section, you made a statement that I think is sweeping and I urge you to be cautious about such statements. ``This study provides a complete overview of the health promotions interventions that are used for the control of hypertension in Africa". You can state that you have made a comprehensive review but not a complete review.

Authors’ Response: The authors agree with the reviewers and have therefore made substantial revisions to the statement.

CONCLUSION

We generally believe that we appropriately incorporated the changes and suggestions made by the reviewers and are positive that this manuscript will meet the criteria for publication in your esteemed journal. 

Yours faithfully,

Kennedy Diema Konlan

---

## [Decision Letter · Decision Letter 1]

10 Nov 2021

Health promotion interventions for the control of hypertension in Africa, a systematic scoping review from 2011 to 2021.

PONE-D-21-28474R1

Dear,

We’re pleased to inform you that your manuscript has been judged scientifically suitable for publication and will be formally accepted for publication once it meets all outstanding technical requirements.

Kind regards,

Muhammad Shahzad Aslam, Ph.D.,M.Phil., Pharm-D

Academic Editor

PLOS ONE

Additional Editor Comments (optional):

Reviewers' comments:

Reviewer's Responses to Questions

**Comments to the Author**

1. If the authors have adequately addressed your comments raised in a previous round of review and you feel that this manuscript is now acceptable for publication, you may indicate that here to bypass the “Comments to the Author” section, enter your conflict of interest statement in the “Confidential to Editor” section, and submit your "Accept" recommendation.

Reviewer #1: All comments have been addressed

Reviewer #2: All comments have been addressed

2. Is the manuscript technically sound, and do the data support the conclusions?

Reviewer #1: Yes

Reviewer #2: Yes

3. Has the statistical analysis been performed appropriately and rigorously? 

Reviewer #1: Yes

Reviewer #2: N/A

4. Have the authors made all data underlying the findings in their manuscript fully available?

Reviewer #1: Yes

Reviewer #2: Yes

5. Is the manuscript presented in an intelligible fashion and written in standard English?

Reviewer #1: Yes

Reviewer #2: Yes

6. Review Comments to the Author

Reviewer #1: (No Response)

Reviewer #2: I would like commend you for addressing all the issues I raised in my first review. I believe this manuscript will be an important guide for health promotion professionals in designing and implementing culturally appropriate interventions to reduce hypertension and its effects on individual, community, and public health.

7. PLOS authors have the option to publish the peer review history of their article (what does this mean?). If published, this will include your full peer review and any attached files.

Reviewer #1: No

Reviewer #2: No

---

## [Editor Report · Acceptance letter]

15 Nov 2021

PONE-D-21-28474R1 

*Health promotion interventions for the control of hypertension in Africa, a systematic scoping review from 2011 to 2021.*

Dear Dr. Konlan:

I'm pleased to inform you that your manuscript has been deemed suitable for publication in PLOS ONE. Congratulations! Your manuscript is now with our production department. 

Kind regards, 

on behalf of

Dr. Muhammad Shahzad Aslam 

Academic Editor

PLOS ONE